# Use of Ground Penetrating Radar in the Evaluation of Wood Structures: A Review

Brunela Pollastrelli Rodrigues [1], Christopher Adam Senalik [2,*], Xi Wu [3] and James Wacker [2]

1    Graduate Program of Forest Science, Southwestern Bahia State University, Bem Querer Drive, Vitoria da Conquista, Bahia State 45083-900, Brazil; brunelafloresta@yahoo.com.br
2    Forest Products Laboratory, Forest Service, USDA, Madison, WI 53726, USA; james.p.wacker@usda.gov
3    School of IoT Engineering, Jiangnan University, Wuxi 214122, China; 7161905006@vip.jiangnan.edu.cn
*    Correspondence: christopher.a.senalik@usda.gov; Tel.: +1-608-231-9309

**Abstract:** This paper is a review of published studies involving the use of ground penetrating radar (GPR) on wood structures. It also contains background information to help the reader understand how GPR functions. The use of GPR on wood structures began to grow in popularity at the turn of the millennium. GPR has many characteristics that make it attractive as an inspection tool for wood: it is faster than many acoustic and stress wave techniques; it does not require the use of a couplant; while it can also detect the presence of moisture. Moisture detection is of prime concern, and several researchers have labored to measure internal moisture using GPR. While there have been several laboratory studies involving the use of GPR on wood, its use as an inspection tool on large wood structures has been limited. This review identified knowledge gaps that need to be addressed to improve the efficacy of GPR as a reliable inspection tool of wood structure. Chief among these gaps, is the ability to distinguish the type of internal feature from the GPR output and the ability to identify internal decay.

**Keywords:** ground penetrating radar; GPR; wood; nondestructive testing; inspection

## 1. Introduction

Ground penetrating radar (GPR) is a nondestructive inspection tool based upon the electromagnetic (EM) theory that radio wave propagation is governed by the EM properties of a dielectric material [1]. It can be used to probe any low-loss dielectric material such as concrete, asphalt, and wood [2]. There are several characteristics, described below, which make GPR an attractive nondestructive evaluation (NDE) tool for wood. The basic GPR unit is comprised of three components: a transmitting and receiving antenna(s), an EM pulse generator, and a data acquisition system. GPR inspection can be performed either through an object or from one side of the object. For inspection through the object, the transmitting and receiving antennas are placed on opposite sides of the object; the transmitting antenna directs the radiation through the object to a receiving antenna. For single-sided inspection, the transmitting and receiving antennas are on the same side of the object; it is common for the transmitting and receiving antennas to be housed within the same device. The transmitter directs waves into the object, the wave energy is scattered or reflected back toward the transmitter/receiver, and the receiver records the reflected waves as raw data for processing and interpretation [3]. This paper is a review of published studies involving the use of ground penetrating radar (GPR) on wood structures. It also contains background information to help the reader understand how GPR functions.

### 1.1. Uses and Advantages of GPR

Ground penetrating radar has several characteristics which make it an attractive inspection tool for wood [1,4–9]. One of the strongest attributes of GPR is the rapid speed of inspection. During an inspection using GPR, the antenna is moved across the surface

of the inspected object/structure. The distance between scans is configurable by the user, but a scan every 3 mm is not uncommon, effectively conducting full field scanning, as opposed to a point by point data collection method which requires interpolation to derive a full field data set. During field inspection, the authors moved the antenna by hand at a rate of 0.2 ms$^{-1}$ and obtained sharp output data. The width of the scan is governed by the size of the antenna. Therefore, assuming a 76 mm scan width for a handheld antenna, an area of 550 m$^2$ could be scanned within an hour. An area the size of an American football field could be scanned by hand in under 10 h. The speed of inspection can be increased by affixing the unit to a carriage, allowing for scanning to be performed at walking speed. The speed of inspection is much faster than many acoustic or stress wave techniques, which require point by point inspection. The ability to scan large areas rapidly is important for structures such as timber bridges, where time out of service is highly disruptive. Another advantage is the ability to estimate feature depth in addition to location. Data collected can be displayed in 1D, 2D, or 3D images. Several adjacent 1D scans can be combined to provide a 2D image, referred to as a radargram, of the internal structure of the inspected object. Several radargrams can be combined to provide a 3D image. Features that are visible include internal moisture pockets, knots, voids, and metal connectors. The ability to detect the presence of internal moisture is important because the presence of moisture is often associated with—or as a precursor to—the development of decay. Lastly, the display of the output radargram allows for even inexperienced users to locate internal features, though the type of feature may not be easily distinguished.

Figure 1 shows the use of GPR on a 2 × 4 over an aluminum sheet. Figure 2 shows how a series of 1D scans is used to construct a 2D radargram. Figure 3 shows three internal features as they appear in a GPR output radargram. The features are numbered one through to three; Features 1 and 2 are internal knots; Feature 3 is a machined circular void. While all three features are clearly visible and easily located, identifying the nature of the features is more difficult.

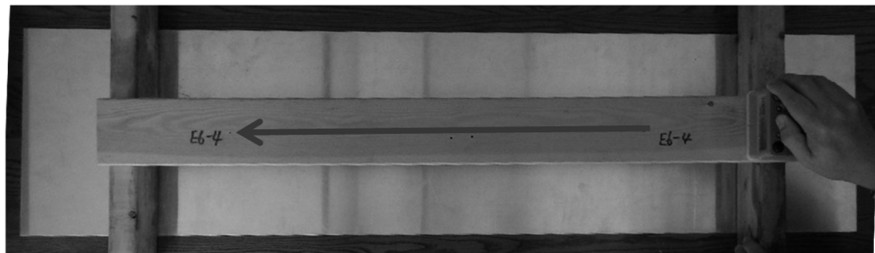

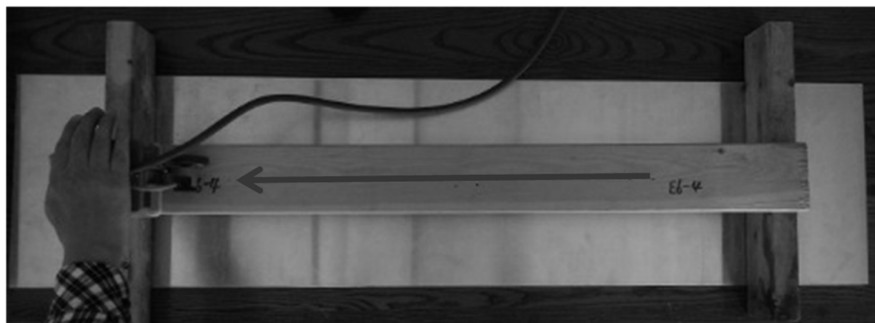

**Figure 1.** Diagram showing use of ground penetrating radar (GPR) to inspect a 2 × 4 (3.8 × 8.9 cm) over an aluminum sheet.

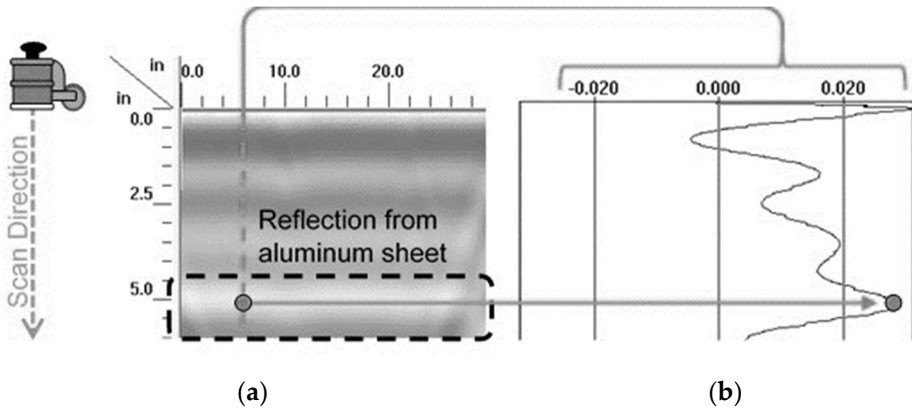

**(a)**                                                                                              **(b)**

**Figure 2.** Typical GPR output for 2 × 4 inspection. (**a**) Radargram is a "top-down" view of all GPR scans. The light band is a peak caused by the radar-wave reflection from the aluminum sheet. The aluminum sheet is. Inches (12.7-cm) below the antenna. (**b**) A single GPR scan from the radargram (along the dashed line) showing the location of the reflection from the aluminum sheet.

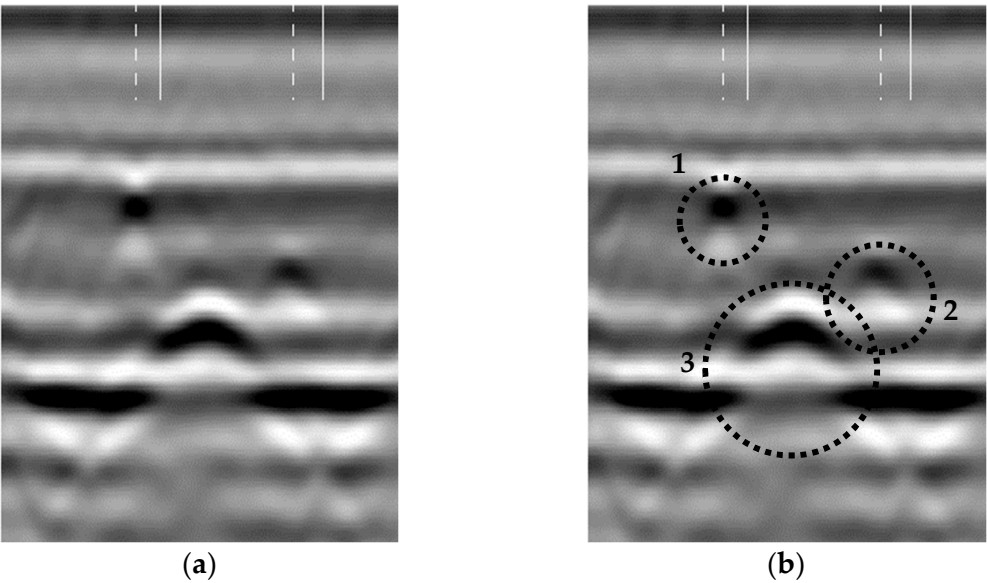

**(a)**                                                                                              **(b)**

**Figure 3.** Typical radargram produced by GPR. Image (**a**) shows the radargram ash shown on the GPR display. Image (**b**) shows three internal features circled in dotted lines. Features 1 and 2 are internal knots. Feature 3 is a circular void that was drilled into the wood.

### 1.2. Limitations and Recommendations for Use of GPR

GPR, as an inspection tool, has several limitations. The propagation of radar through wood is affected by the moisture content, density, temperature, shape and size of the inspected object, shape and size of the internal feature, and preservative treatments [1,8,10,11]. It is not uncommon for several of these factors to be present simultaneously, which can complicate data interpretation. While the location of internal features is intuitive, identifying the nature of a located defect (knot, void, metal connected, etc.) requires an experienced technician [5]. GPR systems have several user configurable settings including gain and frequency pass filters. The ability of the inspector to locate and identify internal features is greatly affected by these settings. Overly high gain can make otherwise inconsequential features appear large (false positive); insufficient gain can diminish relevant features causing them to be overlooked (false negative). Frequency range of the radar wave affects penetration depth and resolution. As frequency decreases, penetration depth tends to increase, but the minimum size defect that is detectable increases. Conversely, as frequency

increases, the size of the detectable defect decreases, but penetration depth tends to decrease [6]. Compensating for the loss of penetration depth using gain can lead to false positive errors, as described above. The configuration of the settings must account for the shape and size of the inspected object. Knowledge of how the settings affect the GPR output can be taught in a classroom setting, but the understanding necessary to properly apply that knowledge in the field comes with experience.

## 2. Dielectric Properties of Wood

Wood is a heterogeneous, anisotropic, and dielectric material. The electrical properties of wood depend principally on moisture content, temperature, density, species, and type of wood (e.g., sapwood versus heartwood), as well as on the fiber orientation (grain) with respect to the electric field direction [11,12]. Moisture content can be considered as one of the most important parameters that affect the EM properties of wood, and thus, the propagation of the EM waves [13,14]. Several authors have documented how moisture content impacts wave propagation, as well as other variables previously noted [1,10,11,15–18].

### 2.1. Permittivity, Dielectric Constant, and Loss Factor

The ability of GPR to locate internal features depends upon the dielectric properties of the wood and the feature. Just like mechanical properties, dielectric properties of wood are cylindrically orthotropic. The dielectric properties of wood are distinguished in the three directions that are generally adopted for examination of wood structure, namely, the longitudinal (L), radial (R), and tangential (T) directions [10]. Figure 4 shows the three principal directions of wood. Local variations in wood density and moisture content influence dielectric properties [19].

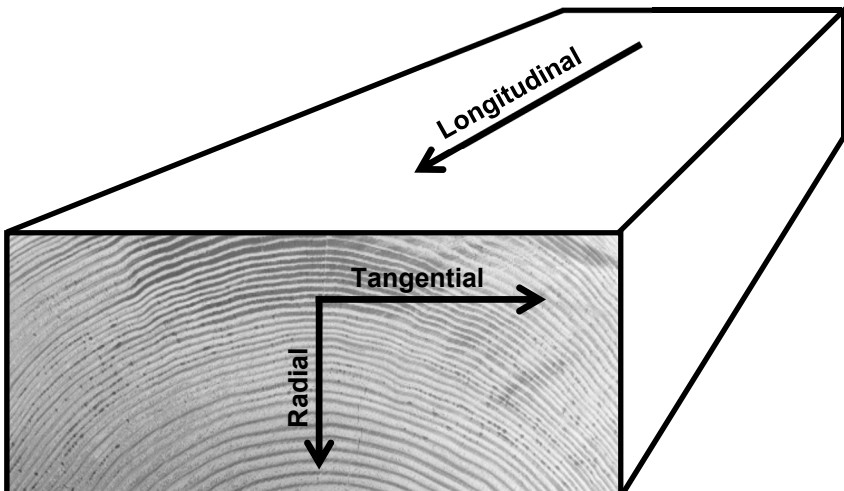

**Figure 4.** Wood principal axes with respect to grain direction and growth rings: longitudinal (L), radial (R), and tangential (T).

Permittivity describes the ability of a material to store and release electromagnetic (EM) energy in the form of electric charge [3]. Permittivity is normally represented as a frequency-dependent imaginary number. The real component of the number represents the capacity of the material to instantaneously store and release energy. The imaginary component represents the energy dissipation or loss of the material. Often, the permittivity of a material is expressed as a nondimensional ratio comparing the permittivity of the material to that of a vacuum. This ratio is referred to as the relative permittivity, $\varepsilon_r$, and is shown in Equation (1).

$$\varepsilon_r = \frac{\varepsilon}{\varepsilon_0} = \frac{\text{Permittivity of material }(\varepsilon)}{\text{Permittivity of free space or vacuum }(\varepsilon_0)} \tag{1}$$

It is in the context of this ratio that the term "dielectric constant" is seen. Dielectric constant (DC), represented by $\varepsilon'$, refers to the real component of the relative permittivity, and the dielectric loss tangent (tg$\delta$) is the imaginary component [10]. The DC of wood is highest parallel to the wood grain. The energy absorbed by the wood is transformed into thermal energy. The complex representation of the relative permittivity is shown in Equation (2).

$$\varepsilon_r = \varepsilon' - i\varepsilon'' = \varepsilon'(1 - i\mathrm{tg}\delta) \tag{2}$$

where $\varepsilon'$ is the relative dielectric constant (real component), $i$ is the imaginary number $\sqrt{-1}$, $\varepsilon''$ is the loss factor (imaginary component), tg$\delta$ is the loss tangent, tg$\delta = \varepsilon'' / \varepsilon'$.

The antenna emits EM waves, which travel along a path through the inspected object. Changes in the DC along the ray path of the EM wave cause reflections, which are detected by the antenna. If the inspected object has the same DC throughout, then no discernable reflections are created. Large changes in DCs in contiguous materials cause the most discernable reflections. The DC of air is close to 1; for water at 20 °C, the DC is close to 80 [20]. The DC for dry wood (ignoring extractives) changes based upon the frequency of the EM wave used and the temperature. For the gigahertz range at 20 °C and moisture content below the fiber saturation point, the DC is less than 4 [10].

### 2.2. Measurements of Dielectric Properties

This section summarizes the findings of several studies that measured dielectric properties of wood. Tables 1 and 2 focus on the dielectric properties of Douglas Fir. Table 2 summarizes the findings of several studies involving a variety of species. A standard for determining dielectric constant and the loss tangent of wood is described in ASTM D150 [21]. The standard describes methods used to determine relative permittivity, dissipation factor, loss index, power factor, phase angle, and loss angle of specimens of solid electrical insulating materials for a frequency range between less than 1 Hz to several hundred megahertz. ASTM D5568 [22] describes how to measure the relative complex permittivity and relative magnetic permeability of solid materials using waveguides for frequencies within the range of 100 to 30 GHz. For a nonmagnetic material, such as wood, the methods described are acceptable to measure permittivity only.

The permittivity of wood depends on several parameters including frequency, voltage, deterioration, moisture content, oven-dry density, temperature, and electrical field orientation relative to the fiber direction [1,23]. There are two basic techniques for dielectric measurements: transmission and reflection methods. In the transmission method, the dielectric material is located between the transmitting and receiving antennas. In the reflection method, the transmitting and receiving antennas are both located on same side of the dielectric material.

James and Hamill [15] measured the dielectric properties of Douglas-fir using a commercially available microwave dielectrometer. They used a klystron (a high-powered microwave vacuum tube that acts as an amplifier) microwave generator and a slotted section of waveguide. The frequencies used in that study were 1, 3, and 8.53 GHz. The dielectric properties were determined for each combination of growth ring orientation, moisture content, and frequency. The authors reported that the DC at all frequencies increased with increasing moisture content and the DC parallel to grain was larger than that transverse to grain at the moisture and frequency combinations evaluated. The average values are shown in Table 1.

Data comparable with those shown in Table 1 were produced by James et al. [19] and are given in Table 2. In that study, the microwave frequency used was 4.81 GHz. A 2D tensor was used to estimate the moisture content, density, specimen thickness, grain angle, DC, and loss tangent. Wood moisture content was found to correlate well with both the DC and loss factor; however, the loss tangent, the ratio of DC to loss factor, did not correlate as clearly as the two components separately.

**Table 1.** Douglas-fir transverse dielectric constant and loss tangent by moisture content.

| Nominal Percentage Moisture Content [1] (%MC) | Frequency 1.0 GHz | | Frequency 3.0 GHz | |
|---|---|---|---|---|
| | Dielectric Constant (DC) | Loss Tangent (tgδ) | Dielectric Constant (DC) | Loss Tangent (tgδ) |
| 7 | 1.9 | 0.05 | 1.8 | 0.06 |
| 10 | 2.2 | 0.10 | 2.0 | 0.11 |
| 12 | 2.2 | 0.13 | 2.0 | 0.12 |
| 16 | 2.6 | 0.13 | 2.4 | 0.15 |
| 22 | 3.4 | 0.17 | 2.9 | 0.17 |
| 25 | 4.0 | 0.17 | 3.3 | 0.21 |
| Green (75) | 10.2 | 0.16 | 9.8 | 0.18 |

[1] Percentage of oven-dry weight. Moisture content shown is the nearest whole number to the average moisture of the individual specimens [15].

**Table 2.** Comparison of dielectric constant of Douglas-fir for two studies.

| Grain Direction | Quantity | James and Hamill [15] | James et al. [19] |
|---|---|---|---|
| | Moisture Content 6% | | |
| Parallel | Dielectric constant | 2.35 | 2.0 |
| | Loss tangent | 0.064 | 0.13 |
| Perpendicular | Dielectric constant | 1.9 | 1.8 |
| | Loss tangent | 0.050 | 0.0065 |
| | Moisture Content 12% | | |
| Parallel | Dielectric constant | 2.73 | 2.85 |
| | Loss tangent | 0.26 | 0.22 |
| Perpendicular | Dielectric constant | 2.16 | 2.0 |
| | Loss tangent | 0.14 | 0.12 |

Sahin and Ay [24] worked with three widely used hardwood species in Turkey and summarized the results of dielectric measurements conducted at two frequencies (2.45 and 9.8 GHz). In that study, a slotted waveguide and standing wave ratio (SWR) meter was used to determine dielectric properties.

Mai et al. [1] measured the permittivity of wood using the weak perturbation method (WPM). The authors used spruce and pine specimens with dimensions of 165 by 82 by 514 mm. The measurements were made at room temperature (~20 °C), and the moisture content of samples was measured by weighing the sample. The results showed that the permittivity measured was affected by moisture, fiber direction (parallel was greater than at perpendicular to fiber direction), and wood oven-dry density (difference between spruce and pine). The results of the study showed that WPM indicated a piecewise linear dependence of permittivity to moisture content with change of slope at the fiber saturation point. The DCs determined using WPM were comparable with those determined using GPR for the same frequencies. The disadvantage of the WPM method was the limited range of frequencies across which measurements could be taken.

Table 3 lists research studies that used microwave frequencies to determine the dielectric properties based on moisture content.

**Table 3.** Research using microwave frequencies to determine dielectric properties (DP) based upon wood moisture content (MC).

| Reference | Species/Material | Equipment/Method | Frequency | MC (%) | Findings |
|---|---|---|---|---|---|
| Mai et al. [1] | Spruce (*Picea abies* (L.) Karst.), Additionally, Pine wood | 1.26 (using resonance-WPM): Vector Network Analyzer (ANRITSU 37325A) | —— | 12, 20 | $\varepsilon'$ at 12% = 2.5 $\varepsilon'$ at 20% = 4.0 |
| Torgovnikov [10] | Softwoods | —— | 3 GHz | 35% | $\varepsilon'$ 1.5 |
| James and Hamill [15] | Douglas-fir | Commercial microwave dielectrometer. klystron microwave generators and a slotted section of waveguide. | 1, 3, and 8.53 GHz | 6, 12 | Table 1 |
| James et al. [19] | Douglas-fir | 20 different electrode forms | 0.02, 0.1, 1, 10, and 100 kHz | 6.5, 12.5, 17, 20.5 | Table 2 |
| Sahin and Ay [24] | *Populus x euramericana* | Means of a slotted waveguide and standing wave ratio meter | 2.45 GHz | 0% to 28% | $\varepsilon'^T$: 1.84 $\varepsilon'^R$ 2.03 $\varepsilon'^\perp$ 1.94 $\varepsilon'^{//}$ 2.49 |
| | *Alnus glutinosa* subsp. Barbata | | | | $\varepsilon'^T$ 2.17 $\varepsilon'^R$ 2.37 $\varepsilon'^\perp$ 2.27 $\varepsilon'^{//}$ 2.87 |
| | *Fagus orientalis* Lipsky | | | | $\varepsilon'^T$ 2.60 $\varepsilon'^R$ 2.81 $\varepsilon'^\perp$ 2.71 $\varepsilon'^{//}$ 3.36 |
| | *Populus x euramericana* | Means of a slotted waveguide and standing wave ratio meter | 9.8 GHz | 0% to 28% | $\varepsilon'^T$ 1.72 $\varepsilon'^R$ 1.85 $\varepsilon'^\perp$ 1.79 $\varepsilon'^{//}$ 2.28 |
| | *Alnus glutinosa* subsp. Barbata | | | | $\varepsilon'^T$ 1.96 $\varepsilon'^R$ 2.06 $\varepsilon'^\perp$ 2.01 $\varepsilon'^{//}$ 2.49 |
| | *Fagus orientalis* Lipsky | | | | $\varepsilon'^T$ 2.36 $\varepsilon'^R$ 2.53 $\varepsilon'^\perp$ 2.45 $\varepsilon'^{//}$ 3.05 |
| Peyskens et al. [25] | European pine, Spruce, Hemlock. | Slotted waveguide, standing wave ratio meter, DP for 3 wood principal axes | 3 GHz | 3% to 35% | DC and tg$\delta$ // 2 to 3× higher $\perp$ to grain. DP varies by species. |

The dielectric constant symbols $\varepsilon'^T$, $\varepsilon'^R$, $\varepsilon'^\perp$, and $\varepsilon'^{//}$ are Tangential, Radial, Perpendicular to grain, and Parallel to grain, respectively

## 3. Applications of GPR on Wood Material

This section reviews several studies involving the use of GPR on wood structures. In addition, the findings of many of the researchers are compiled in Table 4.

Muller [5,6] used a variety of inspection techniques to evaluate the defect prediction capability of GPR for wood structures. Muller [5] examined four salvaged round wood girders from the Purga Creek bridge in Queensland, Australia, during demolition. The girders were rough hewn and round members. Possible defect locations identified using GPR were tested using drilling inspection, ultrasound techniques, and ultimately, destructive testing, in which the girders were cut at identified defect locations. In another study by Muller [6], the same inspection technique was used to examine the timber girders of the Redbank Creek bridge, also in Queensland, Australia. Excellent agreement was found between GPR-predicted defect locations and the drilling survey. The author envisioned using GPR as a first pass inspection tool to locate problem areas for targeted, heightened evaluation in order to reduce the need for drilling inspections.

**Table 4.** List of the studies using GPR technique for wood and wood materials.

| Reference | Species/Material | Equip-ment | Freq. (GHz) | %MC | Results |
|---|---|---|---|---|---|
| Mai et al. [1] | Spruce (*Picea abies*) and Pine wood | GSSI SIR 3000 | 1.5 | 0, 5, 10, 15, 20, 25, and 30 | For each wood type, three samples were tested at different MC (by mass): from 0% to 30%. |
| Muller [5,6] | Timber bridge girders after demolition | GSSI SIR-2000 | 1.2 | — | Gamma ray transmission and ultrasound techniques. Demonstrated GPR ability to detect defects [5]. High correlation between GPR features, test drilling, and postmortem girder cutting. Permittivity 6.4 to 7.4 [6]. |
| Rodríguez-Abad et al. [7] | 22 sawn timbers with side widths 14 to 44 cm | GSSI SIR-10H | 1.5 | Dried and humidified | Maximum perception depth of 44 cm. Antenna placed on the surface of the sample and the air 35 cm offset. |
| Hans et al. [8,9,18] | Black spruce, Quaking aspen, Balsam poplar | TR1000 | 1 | — | MC estimates using GPR: early time signal [8], partial linear regression of signal [9], and GPR wave vel. [18] |
| Redman et al. [11] | Aspen (*Populus sp.*) | TR1000 | 1 | 29, 31, 32 | Size effects on MC measurements of logs using GPR |
| Rodríguez-Abad et al. [26] | *Pinus pinaster* Ait. | GSSI SIR-10H | 1.6 | 12 | $v^{//}EM = 21.5$ cm/ns $\varepsilon'^{//} = 2.0$ $v^{\perp}EM = 21.8$ cm/ns $\varepsilon'^{\perp} = 1.9$ |
| Lorenzo et al. [27] | Maritime pine (*Pinus pinaster*) | GSSI SIR-10 | 0.2, 0.5, 0.9, 1 | Dry | Velocity $11.3 \pm 1$ cm/ns. Average relative dielectric permittivity approximately 7.0 |
| Brashaw [28] | Southern-yellow-pine screw lam. deck w/bituminous top layer | IDS Georadar Aladdin | 2 | 8 to 15 | GPR finds large defects. Signal interpretation complex. Affected by steel. Damped by bituminous layer. |
| Wacker et al. [29] | Douglas-fir (*Pseudotsuga menziesii* (Mirb.) Franco) | GSSI SIR 4000 | 2.5 | 12% Equi-librium moisture content (EMC) w/moisture pockets | Examined several advanced NDE techniques on wood |
| Senalik et al. [30] | Douglas-fir (*Pseudotsuga menziesii*) | GSSI SIR 4000 | 2.5 | Green | Assessed the ability of GPR to locate internal fungal decay in timber bridge sized wood members |
| Wu et al. [31] | Douglas-fir (*Pseudotsuga menziesii*) | GSSI SIR 4000 | 2.5 | 12% EMC w/ moisture pockets | Identified features with the GPR signal and then applied identification on timber bridge girders from Route 66. |

Rodríguez Abad et al. [7] applied the GPR technique to evaluate the effects of density and moisture content on propagation velocity and amplitude of the electromagnetic signal. Two techniques were used for the density analysis, direct contact of the antenna to the specimen, and maintaining a 35 cm air gap between the antenna and the specimen. The air gap method was used to avoid superposition of the direct wave and first reflection on the surface of the specimen. Specimen density ranged from 0.5 to just under 1 g·cm$^{-3}$. Both procedures yielded low correlation between wave velocity and density, with a coefficient of determination, $r^2$, value of 0.47. Wave amplitude was found to relate more strongly to specimen density than wave velocity.

Torgovnikov [10] presented a comprehensive review about dielectric properties of wood in his book. The book examines the effect of moisture, temperature, anisotropy, and wood treatments upon wave behavior. It presents fundamental ideas for EM wave propagation, while specifically focusing upon the specific conditions often encountered while inspecting wood and wood-based materials.

Rodríguez Abad et al. [16,26] presented results showing that wave amplitude had a high correlation with moisture content. The correlation varied by orientation of the antenna relative to the principal directions of the wood. The lowest correlation was found when the antenna was placed such that both the wave propagation direction and the oscillating electromagnetic field were perpendicular to the wood fiber of the specimen. In that orientation, the coefficient of determination between moisture content and normalized amplitude was $r^2 = 0.74$. The signal amplitude was normalized by dividing the signal by the magnitude of the first positive peak while emitting into the air. When the antenna was placed on the side of the specimen, such that the propagation direction was perpendicular to the wood grain but the field oscillated parallel to the wood grain, the value increased to $r^2 = 0.87$. Finally, when the antenna was placed on the end of the beam, such that the propagation direction was parallel to the wood grain and the oscillation direction was perpendicular to the wood grain, the value increased further to $r^2 = 0.93$.

Lorenzo et al. [27] used GPR to evaluate trees, lumber, and tree root systems. A 900 MHz and 1 GHz shielded antenna was used to carry out the analysis on a variety of tree trunks as well as dry timber. Wave propagation velocity was found to be higher in wood structural timber than living trees. This variation was attributed to the increased water content of living trees compared with dry timber. The results indicated GPR may be a useful tool in determining the health of standing trees by determining the relationship between living wood and dry wood.

Brashaw [28] assessed the potential for using GPR to identify and assess simulated deterioration in longitudinal southern-yellow-pine (SYP) timber deck bridges laminated with steel screws. The author performed an assessment of the GPR wave energy signal using visualization software that was provided with the commercially available GPR unit with a high bandwidth antenna centered at 2 GHz. The radar signal was analyzed in both the longitudinal direction (antenna front to back) and the transverse direction (antenna side to side). Interpretation of the radar signals allowed for the identification of various internal defects present in the deck. Based upon the results of the study, the author concluded that GPR has the potential to identify internal defects in wooden bridge decks before and after a bituminous layer was added. However, both the presence of metal within the decks and the presence of the bituminous layer negatively affected the ability of the GPR signal to discern internal features of wood structural members.

Hans [8,9,18] investigated using GPR to estimate the moisture content of frozen and thawed black spruce, quaking aspen, and balsam poplar. While the wood was frozen, the permittivity of wood had low sensitivity to moisture content; between 1 and 3 for moisture contents ranging from 0 to 100%. When the wood thawed, the permittivity was more sensitive to changes in moisture content. Across the same range of 0 to 100% MC, the permittivity ranged in value from 1 to 11. Two numerical modeling methods of determining MC based upon GPR signal amplitude were investigated. One involved a linear fit between the average envelope amplitude and MC; the other was a partial least square regression between signal amplitude and MC. The partial least square regression yielded better predictions of MC. The accuracy of the model was affected by the species of wood. The models used required significant data input, and models between species led to errors.

Mai et al. [32] studied the sensitivity of GPR to moisture content up to 50% in wood materials. A GSSI SIR 3000 unit GPR (from Geophysical Survey Systems, Nashua, NH) was used with a ground-coupled bow-tie antenna optimized for 1.5 GHz transmission. Specimens examined were spruce and pine wood with dimensions of 20 by 18 by 8 cm. Three measurements were made of each specimen. In each measurement, the antenna was

placed such that the generated field would be parallel to one of the principal directions of the wood specimens: longitudinal, radial, and tangential. The specimens were tested at several moisture contents ranging from oven-dried to a moisture content of 30% by mass. It was found that permittivity for the radial and tangential directions were close in magnitude and GPR had the potential to detect moisture variation in timber members.

Redman et al. [11] proposed to quantify the effect of the shape and size of the wood sample on the accuracy of wood moisture content measurements based on the early time signal amplitude and travel time. The authors reported the numerical modeling results and demonstrated the importance of using realistic shielded antenna models in this application. They demonstrated that GPR sensors are effective in measuring moisture content in wood logs. However, the moisture content measurements based on travel time for antennas mounted perpendicular to the longitudinal axis of the log may be difficult due to difficulties in accurately determining the arrival time of the signal reflected from the end of the log. In conclusion, the authors reported that further work is needed to determine the effectiveness of this approach because it would remove the dependence of the computed electrical properties on log shape and size.

Wacker et al. [29] examined several advanced nondestructive testing techniques for wood including pulse echo mapping, microwave, and GPR. Comparative tests were performed on laboratory-prepared specimens. Some specimens contained voids, whereas others contained pockets of wet sawdust. The inspectors had no knowledge of the internal configuration and contents of the specimens. GPR and microwave were both capable of detecting internal moisture and voids; however, GPR was available in commercially ready product form, and therefore, was further examined in subsequent laboratory studies with lumber and glulam specimens.

Senalik et al. [30] discussed the ability of GPR to detect internal fungal decay within timber-sized Douglas-fir (*Pseudotsuga menziesii*) members. The members were inoculated with *Fomitopsis pinicola* (Sw. Karst), a brown rot fungus that was chosen for its ability to attack Douglas-fir. The untreated wood members were placed outside near Gulfport, Mississippi, and allowed to deteriorate over a period ranging from 6 months to 4 years. It was found that the increased moisture content associated with fungal growth was easily discernable using GPR. The ongoing study is determining whether characteristics of the GPR signal can identify decay within the moisture pockets.

Wu et al. [31] developed processes to identify internal defects based upon characteristics exhibited by the GPR signal. Procedures to identify metal, moisture, and voids were developed to evaluate timber bridge girders that had been in service for over eight decades along historic Route 66, in San Bernardino County, CA. The girders contained cracks, splits, nails, metal bars, voids, and decay. GPR was used to identify the internal features and then validated against nondestructive testing methods commonly used for wood, such as stress wave, visual inspection, and resistance drilling.

## 4. Summary and Needs Assessment

There are several aspects of ground penetrating radar that make it an attractive inspection tool for use on wood and wood structures. GPR is commercially available, portable, does not require the use of a couplant, and is faster than point by point inspection methods. Studies have shown that GPR is capable of detecting moisture pockets, voids, and metal connectors which are critical for assessment of wood structures. Locating internal features using GPR can be accomplished by inexperienced inspectors.

Discontinuities in the dielectric constant in the direction of the wave are detected by GPR. The dielectric constant is the real component of the ratio of the permittivity of radar in the inspected material to the permittivity in a vacuum. DC is frequency dependent and increases with decreasing frequency in wood. Wood below the fiber saturation point has a DC of four or less. As the moisture content within the wood increases, the DC can increase above four to a maximum of 80 for pure water. Similar to strength properties, DC is orthotropic in wood with the highest DC parallel to the wood grain.

Correlating aspects of the radar signal to moisture content has been a focus of many studies. There has been some success in this area; however, the GPR output is also affected by many factors including, but not limited to, grain orientation, temperature, size of inspected object, and density. Given the positive results obtained from research in this area, the development of a GPR based method of moisture content measurement for wood structures at some time in the future is not unreasonable.

The most obvious gap in GPR inspection is the identification of internal features and the location and identification of decay. However, this gap is partially mitigated by the ability of GPR to detect moisture pockets, which are often an indicator of interior decay. As previously stated, an inexperienced inspector using GPR can easily locate internal features within wood structures. Unfortunately, identifying the nature of the feature is more difficult. Knots, voids, and nails produce similar output in GPR radargrams. There is a need for a method by which internal features can be quickly characterized in the field.

There is little research in the area of locating decayed wood with GPR. Currently, inspections using GPR rely upon the presence of moisture as an indicator of decay. However, in the absence of moisture, decay may still be present. There is a need for a method to locate and identify internal decay through characteristics of the GPR signal. Ideally the method will be independent of the presence of moisture. If these two areas of research can be addressed, GPR will be a powerful inspection tool for wood-based structures.

**Author Contributions:** B.P.R. collected and organized the original massive part of pieces of information including the tables; C.A.S. supplemented existing manuscript with additional references, created figures, prepared the original manuscript for submission, provided critical feedback and writing—review and editing; X.W. included current references; J.W. resources, writing—review and editing. All authors have read and agreed to the published version of the manuscript.

**Funding:** This study was conducted under a joint agreement between the Federal Highway Administration (FHWA)—Turner-Fairbank Highway Research Center, and the United States Department of Agriculture, Forest Service—Forest Products Laboratory (FPL). The study is part of the Research, Technology, and Education portion of the National Historic Covered Bridge Preservation (NHCBP) program administered by the FHWA. The NHCBP program includes preservation, rehabilitation, and restoration of covered bridges that are listed or are eligible for listing on the National Register of Historic Places; research for better means of restoring and protecting these bridges; development of educational aids; technology transfer to disseminate information on covered bridges in order to preserve the Nation's cultural heritage.

**Institutional Review Board Statement:** Not applicable.

**Informed Consent Statement:** Not applicable.

**Data Availability Statement:** Not applicable.

**Conflicts of Interest:** The authors declare no conflict of interest.

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
