# Peer review of "Use of Ground Penetrating Radar in the Evaluation of Wood Structures: A Review"

_forests, doi:10.3390/f12040492_

Round 1

Reviewer 1 Report

I agree that a review on this GPR is relevant for the scientific community but I have some suggestions and comments that need to be addressed.

How can GPR diagnose defects on wood components? What type of data does it generate? Answer these question on section 1.1

Be more specific about the need of training people to use this method. Why is necessary training?
Could you show how the data look like to show the reader the importance of training to enable evaluators to interpret the data?
What about data processing, can it be another disadvantage? You can answer this question on Section 1.2

Then you discuss what we can find in the literature about it.
Another important information that I missed was, is this method used by bridges inspectors of any kind, timber bridges for example, and what about the building and construction sector?
What can be done to

Abstract
I recommend the authors to rewrite the abstract. I see two problems here that need to be solved; 1) disconnected sentences, and 2) lack of information.

Introduction
Instead of using only the word novice, the authors can use inexperienced, beginner...
L23: What do you mean by without contact? please rephrase it
L39-41: Please reword the following sentence "It is released here for publication as a the breadth of information accumulated..."
L68: Did the authors mean "relatively inexperienced"?

Sections 3 and 4 need careful review. I understand this is a review but it has too much "discussions" and comparisons with other study which makes the narrative a little overwhelming to the reader.

Author Response

RESPONSES TO REVIEWER 1 COMMENTS

I agree that a review on this GPR is relevant for the scientific community but I have some suggestions and comments that need to be addressed.

How can GPR diagnose defects on wood components? What type of data does it generate? Answer these question on section 1.1

Response: Added Figures 1, 2, and 3 that show the output and describe how the output shows the internal features.

Be more specific about the need of training people to use this method. Why is necessary training?

Response:  Added commentary on how the inspector could locate the feature using GPR, but would not be able to easily identify what the feature is.

Could you show how the data look like to show the reader the importance of training to enable evaluators to interpret the data?

            Response:  Addressed this comment with Figure 3.

What about data processing, can it be another disadvantage? You can answer this question on Section 1.2

            Response: Data processing occurs on the screen of the GPR.  This is clarified in the text.

Another important information that I missed was, is this method used by bridges inspectors of any kind, timber bridges for example, and what about the building and construction sector?

            Response: No inspectors are using GPR on timber bridges.  This was clarified in the text.

I recommend the authors to rewrite the abstract. I see two problems here that need to be solved; 1) disconnected sentences, and 2) lack of information.

Response:  Abstract was completely rewritten to reflect identified knowledge gaps identified from the literature review.

Instead of using only the word novice, the authors can use inexperienced, beginner...

            Response: Changed the word novice to inexperienced.

L23: What do you mean by without contact? please rephrase it

            Response: Changed statement to “without couplant”

L39-41: Please reword the following sentence "It is released here for publication as a the breadth of information accumulated..."

            Response: Removed the entire paragraph. It did not add to the overall paper.

L68: Did the authors mean "relatively inexperienced"?

            Response:  Sentence was rewritten to eliminate statement.

Sections 3 and 4 need careful review. I understand this is a review but it has too much "discussions" and comparisons with other study which makes the narrative a little overwhelming to the reader.

Response: Sections 3 and 4 were rewritten into a single section.  It was reorganized to be a more traditional literature review format.

Reviewer 2 Report

Use of Ground Penetrating Radar in the Evaluation of Wood Structures

General comments

The manuscript describes the evaluation of wood materials by using Ground-penetrating radar technic. In my opinion, the manuscript looks more an instructions manual than a scientific article. It is describing how the device work and how needs to be used. Moreover, the document has signs of previous submissions that have not been polished accordingly showing a consistent lack of care when preparing the document for the current journal (e.g. citation style and errors). The article is confusing as the introduction presents the paper as "Review" but the heading above the title says "article". These might be an important report for the wood protection and the tree care field however, this should be presented in more detail avoiding a lot of redundant information throughout as it does not contain novel information but specific case studies. There may be good information in this manuscript, but needs to be reviewed with much greater care and the language revised before it can be fairly reviewed.

Author Response

RESPONSE TO REVIEWER 2 COMMENTS

The manuscript describes the evaluation of wood materials by using Ground-penetrating radar technic. In my opinion, the manuscript looks more an instructions manual than a scientific article.  It is describing how the device work and how needs to be used.

Response: Changed the descriptor of the paper from article to review.  Added the words “: A Review” to the title.  Rewrote the abstract to reflect that this paper is a description of GPR.  I feel that part of this is a misconception on the part of the reviewer regarding the purpose of this paper.

Moreover, the document has signs of previous submissions that have not been polished accordingly showing a consistent lack of care when preparing the document for the current journal (e.g. citation style and errors).

Response: Removed references to outside documents and ongoing projects outside of this review.  As a result, the review is a more stand alone document.  Beyond that, I don’t know what to do.  I have used the template supplied.  The document I received had many broken links to Figures, Tables, and Citations.  I have repaired those, but I don’t know how they were broken in the first place.

The article is confusing as the introduction presents the paper as "Review" but the heading above the title says "article".

Response:  Changed the word article to review.

These might be an important report for the wood protection and the tree care field however, this should be presented in more detail avoiding a lot of redundant information throughout as it does not contain novel information but specific case studies.

Response: Rewrote sections 3 and 4 to eliminate redundancy and make the document more reflective of the format of a traditional literature review.

There may be good information in this manuscript, but needs to be reviewed with much greater care and the language revised before it can be fairly reviewed.

Response:  Abstract, Introduction, and sections 3 and 4 are rewritten.  Additional figures have been added to provide insight into the use of GPR.

Reviewer 3 Report

Overall, the manuscript is well-written and concisely presented.  It can be accepted for publication in its present form.

I have only one comment. Please, verify the text and correct "Error! Reference source not found" by inserting the appropriate reference.

The present paper is well-written and clearly presents a very
interesting topic, relevant for the feature identification and
defect classification using a non-destructive testing technique
to assess the internal condition of wood structures. This study
deals with the object detection method of ground-penetrating radar
(GPR) signals by means of empirical mode decomposition (EMD) and
dynamic time warping (DTW), alongside the evidence of differences
in dielectric constants (DC) along the scan path caused by the
presence of water, metal, or air within the wood. The use of GPR
technology allows a rapid acquisition of data, the output being
viewable by means of high-resolution imaging capabilities.
The conclusions are consistent with the evidence and arguments
presented, GPR providing a valuable contribution to the assessment
of the wood structures.

Author Response

RESPONSE TO REVIEWER 3 COMMENTS

I have only one comment. Please, verify the text and correct "Error! Reference source not found" by inserting the appropriate reference.

            Response: All references checked and repaired.

Round 2

Reviewer 2 Report

Accept